# Mechanical Performance of Metallic Bone Screws Evaluated Using Bone Models

**DOI:** 10.3390/ma13214836

**Published:** 2020-10-29

**Authors:** Yoshimitsu Okazaki, Emiko Hayakawa, Kazumasa Tanahashi, Jun Mori

**Affiliations:** 1Department of Life Science and Biotechnology, National Institute of Advanced Industrial Science and Technology, 1-1 Higashi 1-chome, Tsukuba, Ibaraki 305-8566, Japan; 2TANAC Co., Ltd., 4-24 Moto-Machi, Gifu-city, Gifu 500-8185, Japan; e-hayakawa@k-tanac.co.jp (E.H.); s-tanahashi@k-tanac.co.jp (K.T.); 3Instron Japan Company Limited, 1-8-9 Miyamaedaira, Miyamae-ku, Kawasaki-shi, Kanagawa 216-0006, Japan; Jun_Mori@Instron.com

**Keywords:** metallic screw, mechanical performance, maximum torque, maximum pullout load, bone model, polyurethane foam

## Abstract

To evaluate mechanical performance properties of various types of cortical bone screw, cancellous bone screw, and locking bolt, we conducted torsional breaking and durability tests, screw driving torque tests into bone models, and screw pullout tests (crosshead speed: 10 mm/min) after driving torque tests. The 2° proof and rupture torques of a screw, which were estimated from torque versus rotational angle curves, increased with increasing core diameter of the screw. The durability limit of metallic screws obtained by four-point bending durability tests increased with increasing core diameter. The compressive, tensile, and shear strengths of the bone models used for the mechanical testing of orthopedic devices increased with increasing density of the bone model. The strength and modulus obtained for solid rigid polyurethane foam (SRPF) and cellular rigid polyurethane foam (CRPF) lay on the same straight line. Among the three strengths, the rate of increase in compressive strength with the increase in density was the highest. The maximum torque obtained by screw driving torque tests for up to 8.3 rotations (3000°) into the bone models tended to increase with increasing core diameter. In particular, the maximum torque increased linearly with increasing effective surface area of the screw, as newly defined in this work. The maximum pullout load increased linearly with increasing number of rotations and mechanical strength of the bone model. Screws with low driving torque and high pullout load were considered to have excellent fixation and are a target for development.

## 1. Introduction

The demand for osteosynthesis devices used for fixing fractured bones has been increasing yearly, along with the increase in the number of elderly patients with bone fractures, such as osteoporotic fracture. Four-point bending and compression bending tests of osteosynthesis devices (bone plates, intramedullary nail rods, spinal rods, compression hip screws (CHSs), short femoral nails, and metaphyseal plates) have been carried out to measure their bending stiffness, bending strength, and durability. The bending stiffness of bone plates, intramedullary nails, spinal rods, CHSs, short femoral nails, and metaphyseal plates increases with increasing bending strength [1]. The durability limit of various types of osteosynthesis device linearly increases with increasing bending strength. The relationship (durability limit at 10^6^ cycles) = 0.67 × (bending strength) (N·m) (R^2^ = 0.85) has been obtained by regression [1]. However, the mechanical properties of metallic medical bone screws and the mechanical performance measured using bone models have not been evaluated. Also, there have been few reports on the mechanical performance evaluation of metallic bone screws using a bone model for mechanical tests [2,3]. Therefore, in this study, we focused on the effects of the core diameter on the torsional fracture properties and durability of metallic bone screws used for orthopedic implants. In addition, we examined the relationship between the driving torque and pullout strength of bone screws using bone models for mechanical tests. Furthermore, we investigated the relationship between the density and mechanical properties of bone models. In particular, we also studied the effects of manufacturing conditions on the mechanical properties and density of bone models.

Test methods for the mechanical performance of metallic medical bone screws are standardized in American Society for Testing and Materials (ASTM) F 543 [4] and Japanese Industrial Standards (JIS) T 0311 [5]. Typical metallic medical bone screws such as cortical screws (designed primarily to gain biocortical purchase into cortical bone) and cancellous screws (designed primarily to gain purchase into cancellous bone) are specified in these standards. The torsional, durability, driving torque, and axial pullout tests for medical bone screws using bone models are also specified in these standards.

The bone models (rigid polyurethane foams: RPFs) used for testing orthopedic devices and instruments are standardized in ASTM F 1839 [6]. The requirements of this specification are intended to provide a consistent and uniform material with properties similar to those of human cancellous bone to use as a test medium for various orthopedic devices, such as bone screws. The uniformity and consistent properties of RPF make it an ideal material for the comparative testing of bone screws and other medical devices and instruments. Solid rigid polyurethane foam (SRPF) is used as an alternative test medium for human cancellous bone. Closed SRPF is most commonly used for testing screw pullout, insertion, and stripping torque. This ASTM F 1839 standard defines the relationship between the density (the grade designation refers to the nominal density of the foam in its solid final form in units of kg/m^3^) of the RPF and the required (minimum and maximum values) compressive properties (compressive strength and compressive modulus) and shear properties (shear strength and shear modulus). Ten grades of foam have been defined in ASTM F 1839. Their nominal densities are grade 5, 80.1; grade 10, 160.2; grade 12, 192.2; grade 15, 240.3; grade 20, 320.4; grade 25, 400.5; grade 30, 480.6; grade 35, 560.6; grade 40, 640.8; and grade 50, 800.9 kg/m^3^. Grade 5 designates the nominal value of 5 pounds per cubic foot (5 pcf = 5 × 16.02 kg/m^3^).

On the other hand, cellular rigid polyurethane foam (CRPF) has a cell size that is closer to that of human cancellous bone and is most commonly used for testing subsidence, press-fit devices, and cement augmentation. The effects of density on these mechanical properties of SRPF and CRPF bone models were examined in this study. Testing methods for the compressive and shear properties of RPF are standardized in ASTM D 1621 [7] and ASTM C 273/C273M [8], respectively. Also, the test method for the tensile properties of rigid cellular plastics is standardized in ASTM D 1623 [9] and ASTM D 638 [10]. Thus, there are ASTM standards and also evaluation methods for bone models, but there have been no reports on the mechanical performance of clinically used metallic screws.

In this study, to obtain basic data required for the development of metallic screws with excellent mechanical fixation, torsional breaking, durability, driving torque, and screw pullout tests were performed using commercial screws employed in orthopedic surgery. In particular, we investigated the effects of the core diameter on the torsional properties and durability of metallic bone screws and the effects of screw design parameters such as the pitch, core diameter, and effective surface area on the driving torque. Moreover, we also examined the relationship between the driving torque and pullout strength of bone screws using bone models, as well as the relationship between the density and mechanical (compressive, shear, and tensile) properties of bone models. We believe that the results obtained in this study will be useful for the development of screws that are easy to insert and have excellent mechanical fixation. Our results may also be useful for regulatory approval applications for new screws.

## 2. Experimental Method

### 2.1. Metallic Medical Bone Screws

Figure 1 shows the shapes of the three typical types of screw (cortical, cancellous, and locking bolts) used in this study. The following metallic screws widely used for cortical and cancellous bones were adopted for various mechanical tests. As the metallic materials for orthopedic screws, stainless steel (stainless) [11], 20% cold-worked commercially pure titanium grade 4 (C.P. Ti) [12], and Ti-6Al-4V (Ti-6-4) alloy [13] have been used worldwide. Ti-15Zr-4Nb-4Ta (Ti-Zr) alloy has been developed in Japan as a highly biocompatible alloy for long-term biomedical application [14,15,16,17,18] and it is standardized in JIS T 7401-4 [19].

Table 1 shows the manufacturers, materials, catalog numbers, and dimensions of the metallic screws used in this study. Seven types of mainly self-tapping, fully threaded cortical screws with a main length of 50 mm, five types of cannulated cancellous screws, which were partially threaded screws, locking bolts used in intramedullary femoral nails, and pedicle screws were investigated. The thread and core diameters and pitch of the screws were measured using a capability maturity model (CMM) equipped with an imaging probing system (IM-6120, KEYENCE Corp., Osaka, Japan).

### 2.2. Mechanical Tests of Bone Models

Bone models (TANACBONES (T-bone) and SAWBONES (S-bone)) manufactured by Tanac Co., Ltd. (Gifu, Japan) and Sawbones Co. (Washington, DC, USA) were used for the three types of mechanical tests with various grades of bone models, as shown in Table 2. SRPF and CRPF specimens for comparison were used for bone models. Figure 2 shows the structures of typical SRPF and CRPF bone models (T-bones) with different grades (densities). SRPF and CRPF with the same density have different porosities. SRPF has a compact, porous structure with low porosity, whereas CRPF has a porous structure with high porosity. The effects of density on the compressive, shear, and tensile properties of SRPF and CRPF for T- and S-bones were investigated in this study.

The density of the bone models was measured in accordance with ASTM D 1622 [20]. The compressive, shear, and tensile properties of the bone models were measured in accordance with ASTM D 1621 [7], ASTM C 273/C273M [8], and ASTM D 1623 [9], respectively. Figure 3 shows the jigs used in these three mechanical tests. Shimadzu (Tokyo, Japan) Autograph (maximum 50 kN) and Instron (Kanagawa, Japan) 5982 precision universal mechanical testing machines with mechanical testing software were used for compression tests. A compression test specimen had an area (A) of 50.8 mm (length) × 50.8 mm (width) and a thickness (height) of 25.4 mm (H). The thickness direction was selected as the upward direction of the foam. As shown in Figure 3a, the test specimen was sandwiched between upper and lower metal plates and subjected to a compression test. The compression test was performed at room temperature at a compressive speed of 2.54 mm/min to obtain a load–displacement curve. A displacement meter attached between the upper and lower metal plates was used to measure the compressive displacement of the bone model specimen. When yield occurred and a maximum load point existed, the compressive strength (MPa) was calculated by dividing the maximum load (N) obtained from the load–displacement curve by the cross-sectional area of the specimen (A), as shown in Figure 3b. When yield did not occur when the displacement reached 10% of the original thickness, the displacement corresponding to 10% of the thickness of the bone model block was determined from the load–displacement curve, and the load corresponding to the displacement was divided by the cross-sectional area of the specimen to determine the compressive strength. The compressive modulus (MPa) was calculated using the following equation with the slope (Ac) of the linear part of the load–displacement curve: Compressive modulus (Ec) = Ac × H/A.

Instron 8874 mechanical testing machines with mechanical testing software were used for shear tests, which were performed in accordance with ASTM C 273/C273M [8]. The bone model specimen used in the shear tests was 76.2 mm (length, L) × 25.4 mm (width, b) × 6.35 mm (thickness, t). A specimen was attached to the upper and lower plates of the jig, as shown in Figure 3c. Aluminum (Al) alloy (A2024) or carbon steel (S50C) was used for the plates to attach bone model specimens. A two-liquid mixed type (3M Scotch-weld epoxy adhesive, EC-2216B/A, Saint Paul, MN, USA) was used as the adhesive. The adhesion was performed in accordance with the specifications of the adhesive. The material used for the adhesive plates and the method used to clean their surface were selected in accordance with ASTM D 2651 [21]. The surfaces of both the Al alloy and carbon steel plates were polished with 120- and 220-grit, waterproof emery papers. For bone models up to grade 20, the plate surface was thoroughly cleaned with acetone after polishing, then the adhesive was applied. Then, the adhesive surface was clamped to fix it. Bone models of grade 25 and above were attached after chemical etching as recommended in ASTM D 2651. The shear tests were conducted at room temperature and a shear speed of 0.5 mm/min to obtain a load–displacement curve. The displacement of the bone model was measured using a displacement meter. The displacement at the center of the bone model was measured to determine the shear modulus. The shear strength (MPa) was calculated by dividing the maximum load (N) obtained from the load–displacement curve by the shear cross-sectional area of the specimen (L × b). The shear modulus (MPa) was calculated using the following equation with the slope (Aτ) of the linear part of the load–displacement curve: shear modulus (Eτ) = Aτ × t/(L × b).

Instron 5982 mechanical testing machines were used for tensile tests. The test grip shown in Figure 3d was used for the tensile tests of the bone models. Tensile tests were performed in accordance with ASTM D 1623 [9]. The diameter of the test specimen was 28 mm, the gage length in the parallel part was 12 mm, and the tensile speed was set at 1.0 mm/min. The elongation of the bone model was measured with a noncontact extensometer (Instron, Norwood, MA, USA). The surface of the tensile test specimen was coated with black graphite to make it easier to measure the displacement. The tensile modulus was estimated as the slope of the linear part of the tensile strength–strain curve. Mean values and standard deviations were calculated for at least five specimens in the three types of mechanical test.

### 2.3. Mechanical Performance Tests of Bone Screws

Mechanical performance tests of bone screws were conducted in accordance with JIS T 0311 [5] and JIS T 0312 [22]. Instron 8874 mechanical testing machines with motor control rotation equipment and various types of mechanical testing software were employed for experimental purposes. Figure 4a,f shows schematic illustrations of the torsional breaking and durability tests with bone screws and the driving torque and pullout tests with the bone model in accordance with JIS T 0311 [5]. In the torsional breaking test, screws were set about five threads up from the bone screw fixture subject to the screw length. Screws were also set two or three threads up from the bone screw. The screw was attached to the motor control rotation equipment, which was placed downside of the fixture, then the appropriate screwdriver was kept in the screw head so as not to apply an axial load on it. Torsional torque was constantly applied at a rate of 1 rpm until the screw broke. The torque versus rotational angle curves shown in Figure 4b were obtained by measurement. The torque was measured at a point with a rotation of 2° (2° torsional yield torque) and at the breaking point (maximum torque). To examine the effects of the core diameter and material and the screw design on the 2° torsional yield torque and maximum torque, the following cortical, cancellous, and locking bolt screws were used for the torsional breaking tests as shown in Table 1: Depuy Synthes (Tokyo, Japan), cortical 414-040, 280-990, and 480-990VS, cancellous 217-050 and 417-050, and locking bolt 459-300VS; MDM, cortical 14022-50 and cancellous 14225-50; Stryker (Tokyo, Japan), cortical OR-601050 and 1896-5050S; Zimmer Biomet (Tokyo, Japan), cortical 48-2306-050-01, 48-2319-050-00, and 00-2253-050-42; Mizuho (Tokyo, Japan), cortical 810-34, 810-36, 810-38, 810-48, and 01-810-46; Teijin Nakashima (Okayama, Japan), cortical B30and B35 and cancellous B050.

Figure 4c shows the four-point bending durability test on bone screws in accordance with JIS T 0312 [22]. To investigate the effects of the core diameter and material on the durability limit, the following 11 types of fully threaded cortical screw with a total length of 50 mm were used for the four-point bending durability test: Depuy Synthes, 414-850S and 214-850; MDM, 14022-50; Zimmer Biomet, 48-2319-050-00, 48-2319-050-01, 00-2253-050-55, and 00-2253-050-42; Stryker, OR-601050 and 1896-5050S; Teijin Nakashima, B30 and B35. The durability of each metallic screw was evaluated under the following conditions: (distance between loading rollers):(distance between supporting rollers of 4 mm) = 1:3, sinusoidal waves with a frequency of 3 Hz, and stress ratio R (minimum load/maximum load) = 0.1 for more than 1 × 10^6^ cycles. M–L curves (maximum load vs number of cycles to failure on a logarithmic scale) were plotted for the four-point bending durability test. From the obtained M–L curves, the durability limit, which is the maximum load corresponding to 1 × 10^6^ cycles, was determined.

### 2.4. Driving Torque and Screw Pullout Tests

Driving torque tests of model bones and pullout tests with bone screws were conducted in accordance with JIS T 0311 [5]. Instron 8874 mechanical testing machines were used for the driving torque and screw pullout tests. For the driving torque and screw pullout tests, respectively, shown in Figure 4d,f, alternatives to T- and S-bones were used as materials. To investigate the effects of the density of the bone model, the number of rotations (3 to 10), and the screw design parameters, such as the thread diameter, core diameter, and the effective surface area of the screw, on the maximum driving torque and maximum pullout load, we used S-bone SRPF from grade 5 up to grade 40 and CRPF from grade 7.5 up to grade 20 for driving torque and screw pullout tests. The following 17 types of screws (mainly with a total length of 50 mm) were used: Depuy Synthes, cortical 414-040 and 214-850, and cancellous 417-050 and 217-050; Stryker, OR-601050; Zimmer Biomet, 48-2306-050-01, 48-2319-050-00, and 00-2253-050-42; Mizuho, 01-810-48; Teijin Nakashima, cortical B30 and B35 and cancellous B050; Robert Reid (Tokyo, Japan), 2226-2440, 2226-2840, and 2230-07R; Medtronic (Tokyo, Japan), 86945540 and 86946540. Test media for S-bones were fabricated as 4 × 4 cm squares and clamped under fixation. The screwdriver locked onto the fixation was engaged in the screw head. The screw was driven into the bone medium up to a maximum depth of approximately 30 mm (mainly 17–20 mm) with an optimum axial load range of 20 to 1000 N according to the density of the model bone by applying a torsional torque at a rate of 1, 3, or 5 (mainly 3) rpm. The maximum torque up to 14 rotations was recorded in the driving torque tests. The optimum load range in the driving torque tests was determined by changing the load over a wide range so as to obtain the optimum driving depth according to the density of the bone model.

To measure the relationship between the driving torque and the maximum pullout load with the following 23 metallic screws described in Section 2.1, the measured T- and S-bone media were SRPF grade 15 (nominal density: 240.3) and CRPF 20 pcf (nominal density: 320.4 kg/m^3^): cortical screws, 414-040, 214-850, 48-2319-050-00, 48-2306-050-01, 00-2253-050-42, OR-601050, 1896-5050S, 14022-50, 14224-50, 035A-001-050, and Teijin Nakashima, B30, B35; cancellous screws, 417-050, 217-050, 47-2483-095-60, 14225-50, 005A-340-070, and B050; pedicle screws, 86965540, 86946540, 3306-4540, and 3306-5545; locking bolt, 459.500VS. The screw was driven into the bone medium (4 × 4 cm squares) up to a depth of approximately 20 mm with an axial load range of 30 to 1000 N by applying a torsional torque at a rate of 3 rpm. Classically, there are pretapped and self-tapped methods depending on whether a predrilled hole is made. Considering the mechanical effect of the predrilled hole, we chose the self-tapped method. After the driving tests, pullout tests were carried out at a cross speed of 10 mm/min. Maximum pullout loads were determined from load–displacement curves.

### 2.5. Statistical Analysis

Mean values and standard deviations were calculated for at least three specimens in each mechanical test. Linear regression analysis was performed between the durability limit and core diameter of the screw and between the torsional torque and the effective surface area. Also, the L–N curve, the durability limit of metallic screws, and its standard deviation were calculated with statistical analysis software based on Japan Society of Mechanical Engineers (JSMS)-standard (SD)-06-08 [23].

## 3. Results and Discussion

### 3.1. Mechanical Properties of Metallic Screws

Figure 5 shows the changes in 2° yield torque, rupture torque, and rupture angle with the core diameter of the screws. The error bars represent the standard deviation. Both torque values increased with the core diameter. On the other hand, the rupture angle decreased with increasing core diameter. In the comparison of stainless steels and Ti materials, a slight difference in 2° yield torque and a negligible difference in rupture torque were observed. The cortical screw broke from the fixed part of the screw at the bottom. For many cancellous screws, the cylindrical body was twisted, and the threaded portion did not break. For the locking bolts for intramedullary nails, the failure point was immediately below the screw head, not at the screw groove, as there was no significant difference in form between the screw thread and the groove.

In the strength of material analysis for the twisting of a rod with a cylindrical cross section, the maximum torsional moment T_max_ (N·m) generated at the surface of the screw by the torsional breaking test and the rupture angle (θ_R_) of the screw are expressed as:T_max_ = τ_max_ × Z_p_(1)
θ_R_ (rad) = 32 × T_max_ × L/(π × G × d^4^)(2)
G = 32 × T_max_ × L/(π × θ_R_ × d^4^) = 2 × τ_max_ × L/(θ_R_ × d)(3)
where τ_max_ is the maximum surface shearing stress (MPa), Z_p_ = π/16 × d^3^ is the torsional modulus of the section, which depends on the core diameter (d, m) of the screw, L (m) is the distance between the driver tip and the fixed tip of the screw (see Figure 4a), and G (GPa) is the torsional rigidity [24]. The τ_max_ and G values for cortical, cancellous, and locking bolts obtained by the torsional breaking test are summarized in Table 3. In Table 3, the results obtained with screws made of the same material by various manufacturers are shown as means and standard deviations. In addition, Table 1 shows the literature values of 0.2% proof strength (σ_0.2_) and ultimate tensile strength (σ_UTS_) with specimens cut from osteosynthesis devices obtained by room-temperature tensile tests for comparison [1]. The G values of C.P. Ti and Ti-6-4 are not shown in Table 3 as the screws did not break. For cancellous screws, C.P. Ti and stainless steel tended to have higher torsional strength (τ_2°_ and τ_max_) than cortical screws; however, Ti-6-4 alloy tended to have lower torsional strength. Because the height of cancellous screws is larger than that of cortical screws, the torsional strength of Ti-6-4 alloy is considered to have decreased, owing to its high notch sensitivity [15]. On the other hand, τ_2°_ and τ_max_ of the locking bolt close to the rod-shaped thread (smaller groove) were approximately 60 and 80% of σ_0.2_ and σ_UTS_, respectively. Also, τ_max_ of the Ti-6-4 alloy locking bolt was close to the ultimate shear strength (682 MPa) of Ti-6-4 alloy reported in a handbook on the properties of Ti materials [25]. The (ultimate shear strength)/(ultimate tensile strength) ratio of Ti-6-4 alloy reported in this handbook was 0.65%, which is close to the τ_max_ /σ_UTS_ ratio (0.78%) of the Ti-6-4 locking bolt obtained in this study [25].

Figure 6 shows the fatigue properties of screws obtained from the four-point bending durability test. The durability performance of the metallic screws markedly changed with the core diameter, particularly between 3.0 and 3.8 mm, reaching 800 N at a core diameter of 3.8 mm. This value was twofold that at 3.0 mm. This also implies that the core diameter has a greater effect on the fatigue strength than the material. Figure 7 shows the effects of the core diameter on the durability limit obtained by four-point bending durability tests. The durability limit (D, N) increased in proportion to the core diameter (d) of the screws (D = −1413 + 577 × d, R^2^ = 0.97). Thus, the core diameter strongly affects the durability performance.

The durability limit obtained from the four-point bending durability test is considered using the fatigue strength of the material. The maximum moment M (N·m) generated by the four-point bending durability test is expressed as
M = 1/2 × P_m_ × h = σ_m_ × Z(4)
where P_m_ is the maximum load (N), σ_m_ is the maximum surface stress, and Z = π/32 × d^3^ is the section modulus of the screw, which depends on the cross-sectional core diameter (d) of the screw. Fatigue failure occurred in the four-point bending durability test when the surface stress exceeded the fatigue strength of the material. Table 3 shows mean values (σ_D_) of materials with σ_m_ = M/Z calculated from the durability limit obtained in this study. The σ_D_ values for C.P. Ti, Ti-6-4, and stainless steel were approximately 69, 82, and 70% of the mean values of σ_UTS_, respectively, for osteosynthesis devices reported in the literature [1]. These σ_D_/σ_UTS_ ratios were also close to literature ratios (69, 73, and 79%,) of the fatigue strength at 10^7^ cycles to the ultimate tensile strength for C.P. Ti, Ti-6-4 alloy, and stainless-steel specimens, respectively [15].

### 3.2. Mechanical Properties of Bone Models

Figure 8 shows the relationship between the density and the compressive, tensile, and shear properties of the bone models. The results with the same marks shown in Figure 8a,b were obtained under the same test conditions using the same screw and the same bone model. The error bars represent the standard deviation. The compressive, tensile, and shear strengths of the bone model increased with increasing density. In particular, at a density of grade 20 or higher, the rate of increase for the three strengths markedly increased. It was also found that the strength and modulus obtained for the SRPF and CRPF of the T- and S-bones lay on the same straight line. Also, among the three strengths, it was found that the rate of increase in compressive strength with the increase in density was the highest. In the shear test for densities up to grade 20, the same results were obtained for Al alloy and carbon steel loading plates, so the average and standard deviation obtained with two loading plates are shown in Figure 8. The shear test results for grades 25 to 35 are shown for carbon steel. Shear fracture was observed in the bone model up to grade 25. However, at grade 30 or higher, peeling fracture was observed at the bonding surface between the bone model and the metal. Therefore, the results of the shear tests for grade 40 are not shown. The standard deviation of the modulus markedly increased at grade 30 or higher. For these grades, the fluctuation range of the test results was large, so the mechanical tests were repeated five times or more. The relationship between the density and mechanical properties of the bone model used for testing orthopedic devices is shown in Table 4. On the basis of the relationship shown in Table 4, JIS T 0311 was revised in 2020 [5]. The similar tendency observed in the compressive and tensile properties in Figure 8 was similar to the relationship between the compressive and tensile properties reported for metallic materials such as Ti alloys [15]. On the basis of these results, the compression and shear properties are specified in JIS T 0311 [5].

### 3.3. Screw Driving Torque and Pullout Properties

We examined the effects of the parameters related to the shape of the screw, such as the core diameter, thread diameter, height ((thread diameter – core diameter)/2), pitch, and effective surface area as newly defined in this work, on the driving torque. When the loading load in a driving torque test was small, the screw could not be screwed in. When the load was very large, the driving torque increased because the bone model was screwed despite being damaged. The screwing depth and the number of rotations were measured, and the optimum loading load was determined from the linearity between the screwing depth and the number of rotations. The optimum loading load was determined according to the density of the bone models as follows: Grades 5 and 7.5, 30–50 N; grade 10, 40–80 N; grades 12.5 and 15, 80–200 N; grade 20, 100–400 N; grade 30, 280–750 N; grade 40, 500–950 N. These values reflect the rapid changes in mechanical properties due to the increase in density shown in Figure 8. When the rotation speed was 5 rpm, the driving torque tended to increase, and the driving torque at 3 rpm was stable with little variation. Since it was found that the speed of 3 rpm was the optimum speed for driving tests, we decided to use this speed.

The driving torque initially increased linearly with increasing number of rotations, as shown in Figure 4e. The driving torque also markedly increased with increasing density of the bone model. Also, when the thread diameter increased, the driving torque tended to increase. In particular, the torque fluctuated with increasing number of rotations, and its fluctuation width increased with increasing density. Figure 9 shows the effects of the (a) core diameter and (b) screw pitch on the maximum driving torque for up to 8.3 rotations (3000°). Data in Figure 9 were obtained using the S-bone and 14 types of screws (DePuy Synthes, 414-040 and 214-850; Zimmer Biomet, 48-2319-050-00, 48-2306-051-01, and 00-2253-050-42; Stryker, OR-601050; Robert Reid, 2226-2440, 2226-2840, and 2230-07R; Medtronic, 86945540 and 86946540; Teijin Nakashima, B30 and B35 mm; Mizuho, 810-48). As is clear from Figure 9a, the maximum driving torque increased linearly with increasing core diameter of the screw. The dimensions of the metal bone screws are specified in ISO 9268 [25]. The pitches are categorized into 1, 1.75, 2, 2.5, and 2.75 mm. There are many types of cortical screw with a pitch of 1.75 mm. As shown in Figure 9b, except for the screws with a pitch of 1.75 mm, the maximum driving torque for up to 8.3 rotations (3000°) tended to increase as the pitch increased. This maximum torque also increased with increasing number of threads.

On the basis of these results, we discussed the effect of the screw shape on the driving torque. Figure 10 shows the method of calculating the effective surface area of a screw, which affects the maximum driving torque. When a load is applied from the left-hand side, the friction area of the screw that comes into contact with the bone model is considered to be the effective surface area. From the geometrical relationship, and considering that the thread of a screw is spiral, the effective surface area of one rotation of the screw is given by:(a + b) π × d/cosβ × n(5)
where β is the lead angle of the screw and n is the number of rotations. The lengths and angles of various parts of the screw (b, d, r, α, and β in Figure 10) were measured under an optical microscope to obtain the effective surface area per rotation of the screw. The a is given by a = 2π × r × α/360. Figure 11 shows the relationship between the effective surface area calculated using Equation (1) and the maximum driving torque. Figure 11 shows data obtained using the S-bone and the 17 screws described in Section 2.4. When the effective surface area for both SRPF and CRPF bones was increased, the maximum driving torque increased linearly. It is clear from Figure 11 that grade 15 SRPF and grade 20 CRPF lie on the same straight line. The following equations hold between the maximum driving torque T (N·m) and the effective surface area S (mm^2^) for various densities of the bone model.

Grade 15 SRPF: T = 0.00841S – 0.270, R^2^ = 0.93Grade 20 SRPF: T = 0.0209S – 1.168, R^2^ = 0.95Grade 20 CRPF: T = 0.0101S – 0.591, R^2^ = 0.93

With increasing density of the bone model, the multiple correlation coefficient R^2^ increased and the slope obtained by linear regression analysis increased linearly.

Figure 12 shows the relationship between the maximum pullout load (N) after a screw-in driving torque test, the number of rotations, and the density of the bone model. Data in Figure 12 were obtained using the S-bone and four typical types of screws (DePuy Synthes, 414-040 and 417-050; Stryker, OR-601050; MDM, 14022-50). With increasing mechanical strength of the bone model and number of rotations, the maximum pullout load of the screws increased linearly. The same results were obtained with 10 other screws (DePuy Synthes, 214-805 and 217-050; Zimmer Biomet, 48-2319-050-00, 48-2306-051-01, and 00-2253-050-42; MDM, 14224-50; Teijin Nakashima, B30, B35, and B050; Mizuho, 810-48). Figure 13 shows the maximum pullout load as a function of the maximum driving torque obtained using the T- and S-bones and the 23 metallic screws shown in Section 2.4. The maximum pullout load increased linearly with increasing driving torque in cortical, cancellous, and pedicle screws. The maximum pullout loads of the pedicle and cancellous screws tended to be higher than those of the cortical screws. It was also found that the grade 20 CRPF had better linearity than the grade 15 SRPF. Therefore, grade 20 CRPF is recommended for mechanical performance evaluation in JIS T 0311 [5]. Screws with low driving torque and high pullout load are considered to have excellent fixation and are a target for development. To evaluate the mechanical characteristics of bone models simulating osteoporosis, SRPF grade 5 or CRPF grade 7.5 tends to be used [26]. In this study, a pilot hole was not drilled before screwing into the bone model; however, it is considered that the same tendency is exhibited even when a pilot hole is drilled.

Similar driving torque tests for up to 10 rotations (3600°) and screw pullout (crosshead speed: 10 mm/min) tests with SRPF grades 10, 15, and 20 (S-bone) were performed with the lag screws used in CHSs and short femoral nails. Figure 14 shows the measured results obtained with Ti-6-4 lag screws (length 80 mm) used in short femoral nails (Stryker, Gamma 3 (3060-0080S); DePuy Synthes, PFN 473-080VS; Smith & Nephew (Tokyo, Japan), IMHS 7166-1680) [1]. The depth (approximately 25 mm) of the driving torque corresponded to the screw part of the lag screw. The maximum pullout load increased linearly with increasing driving torque of the lag screws. The same result was obtained with the CHS lag screws (DePuy Synthes, TAV Tubeplate 480-800VS; Zimmer Biomet, Ti-Versa Fx II (48-1201-080-00); Stryker, Omega Plus (3368-5-080); Mizuho (01-800-06)) reported in the literature [1]. These basic data obtained in this study are considered to be useful for the development of screws with excellent fixing properties. They may also be useful for regulatory approval applications for new screws. Screws with low driving torque and high pullout load are considered to have excellent fixation and are a target for future development.

## 4. Conclusions

To obtain basic data required for the development of metallic screws with excellent mechanical fixation, torsional breaking, durability, driving torque, and screw pullout tests were performed using commercial screws in orthopedic surgery. We investigated the effects of the core diameter on the torsional properties and durability of metallic bone screws and the effects of screw design parameters such as the pitch, core diameter, and effective surface area on the driving torque. Moreover, we examined the relationship between the driving torque and pullout strength of bone screws using bone models, as well as the relationship between the density and mechanical (compressive, shear, and tensile) properties of bone models.

(1)The 2° yield torque and rupture torque obtained by the torsional breaking test of metallic screws increased with the core diameter of the screws. Using the maximum torsional moment and rupture angle, we estimated the maximum surface shearing stress (τ_max_) and torsional rigidity (G) for cortical, cancellous, and locking bolt screws.(2)The durability limit of screws increased with increasing core diameter. The maximum surface stress (σ_D_) values of C.P. Ti, Ti-6-4 alloy, and stainless-steel screws calculated from the durability limit were 69, 82, and 70% of the ultimate tensile strength (σ_UTS_) for osteosynthesis devices, respectively.(3)The compressive, tensile, and shear strengths of the bone model increased with increasing density of the bone model. The strength and modulus obtained for SRPF and CRPF lay on the same straight line. Among the three strengths, the rate of increase in compressive strength with the increase in density was the highest.(4)The maximum driving torque increased linearly with increasing effective surface area of the screws.(5)The maximum pullout load of the screws increased linearly with increasing density of the bone model and number of rotations. Screws with low driving torque and high pullout load were considered to have excellent fixation and are a target for development.

## Figures and Tables

**Figure 1 materials-13-04836-f001:**
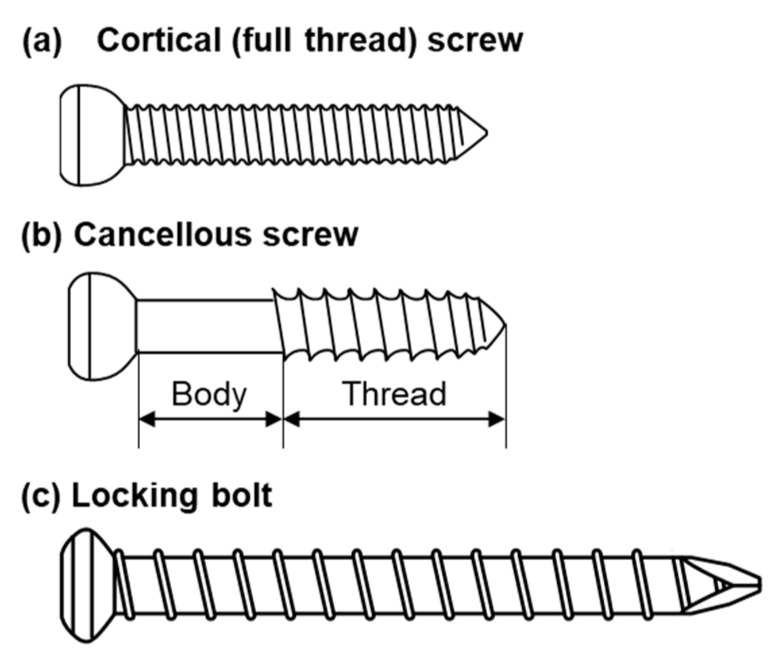
Shapes of typical screws used in this study.

**Figure 2 materials-13-04836-f002:**
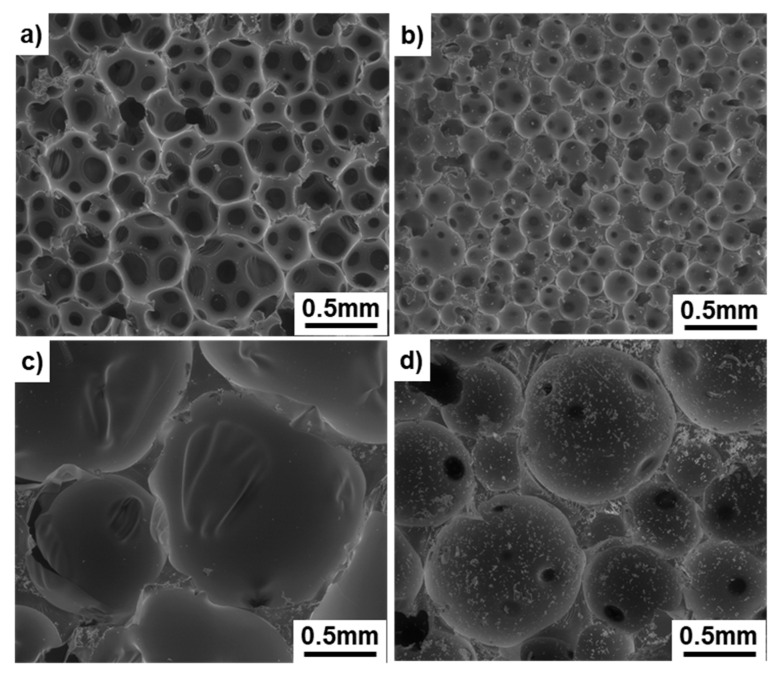
SEM images of solid rigid polyurethane foams (SRPFs) (**a**) T-bone grade 5 and (**b**) grade 15 and cellular rigid polyurethane foams (CRPFs) (**c**) T-bone grade 7.5 and (**d**) grade 20.

**Figure 3 materials-13-04836-f003:**
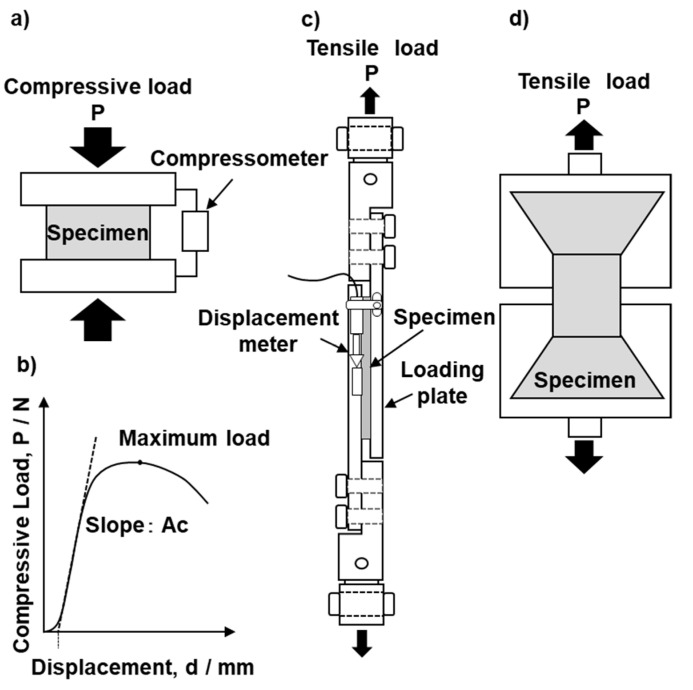
Schematic illustrations of (**a**) compressive test jig, (**b**) compressive load–displacement response, (**c**) tensile plate shear test setup, and (**d**) tensile test grip.

**Figure 4 materials-13-04836-f004:**
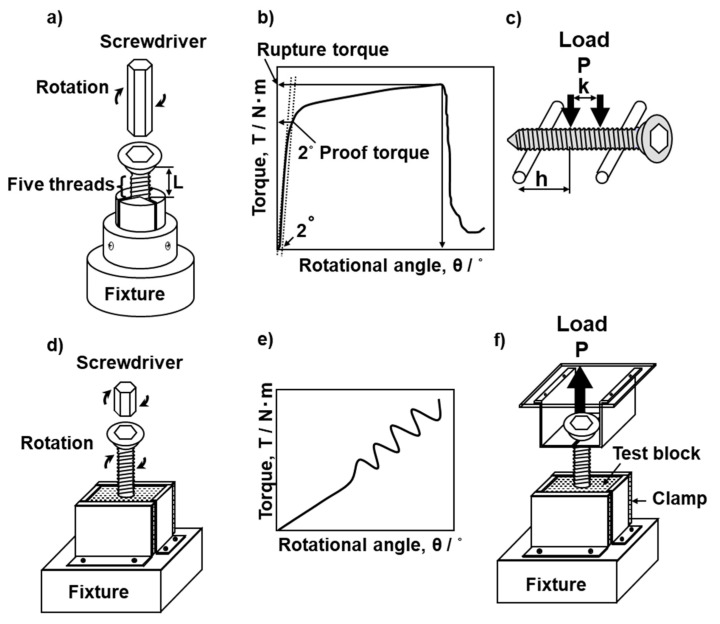
Schematic illustrations of (**a**) torsional breaking test jig of screw, (**b**) torque–rotational angle response, (**c**) four-point bending durability test, (**d**) driving torque jig, (**e**) torque–rotational angle response for bone model, and (**f**) screw pullout test jig.

**Figure 5 materials-13-04836-f005:**
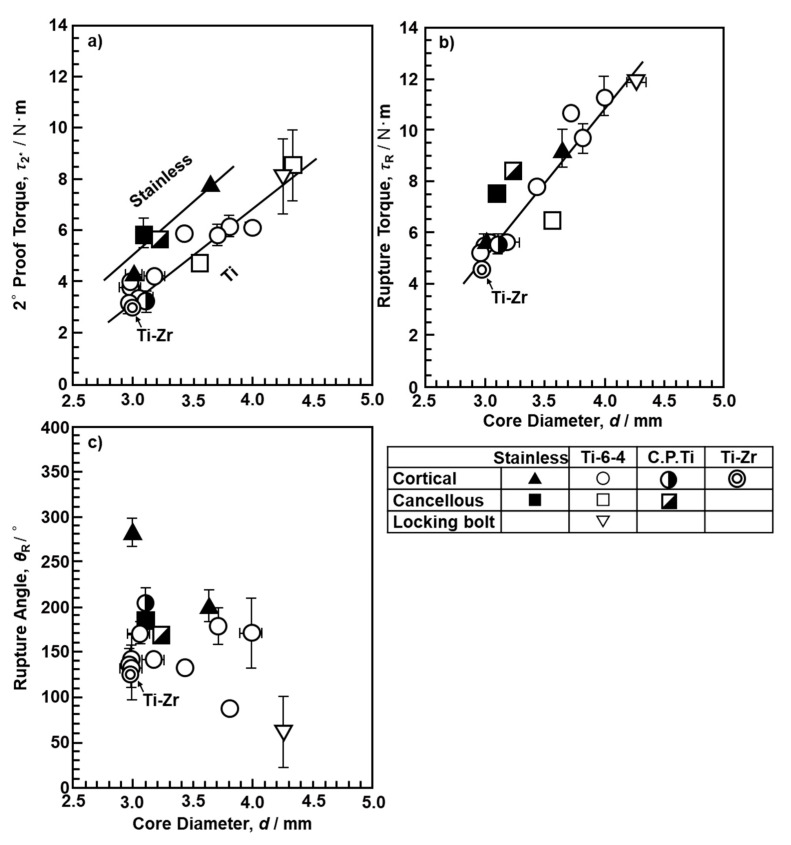
Effects of core diameter of screw on (**a**) 2° proof torque, (**b**) rupture torque, and (**c**) rupture angle.

**Figure 6 materials-13-04836-f006:**
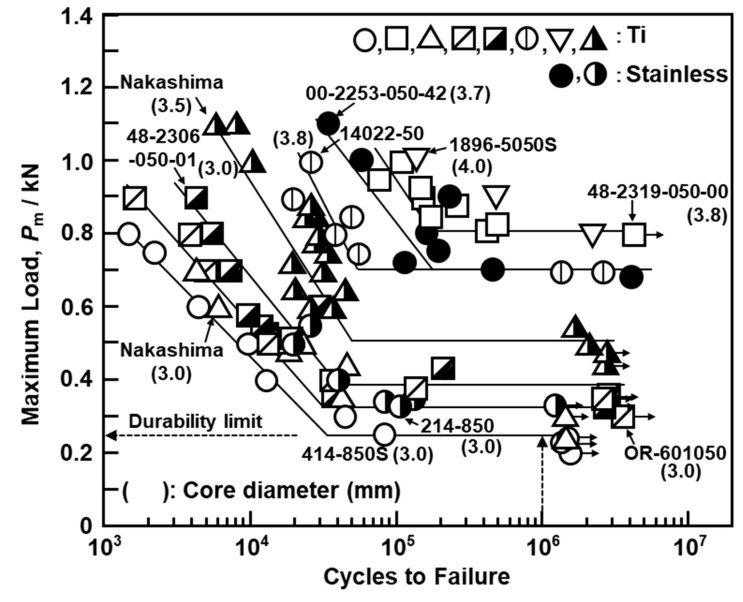
Fatigue properties of screws obtained from four-point bending durability test.

**Figure 7 materials-13-04836-f007:**
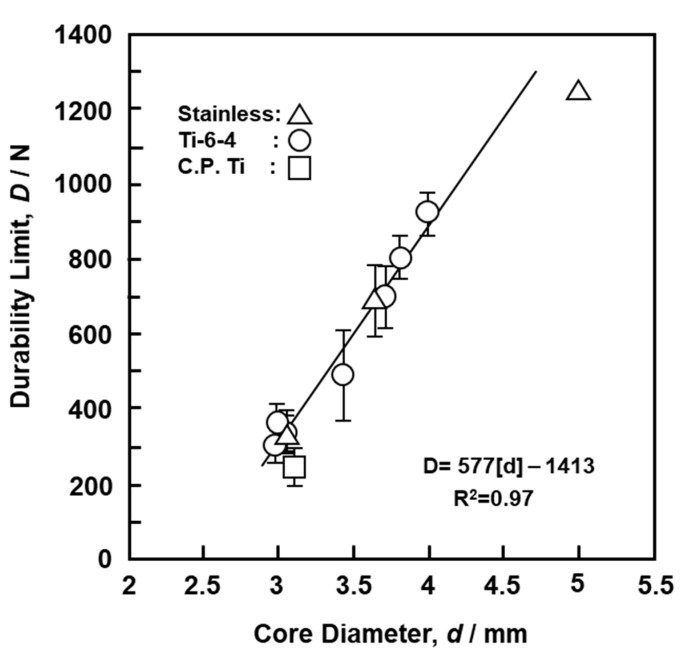
Effects of core diameter on durability limit obtained by four-point bending durability tests with metallic screws.

**Figure 8 materials-13-04836-f008:**
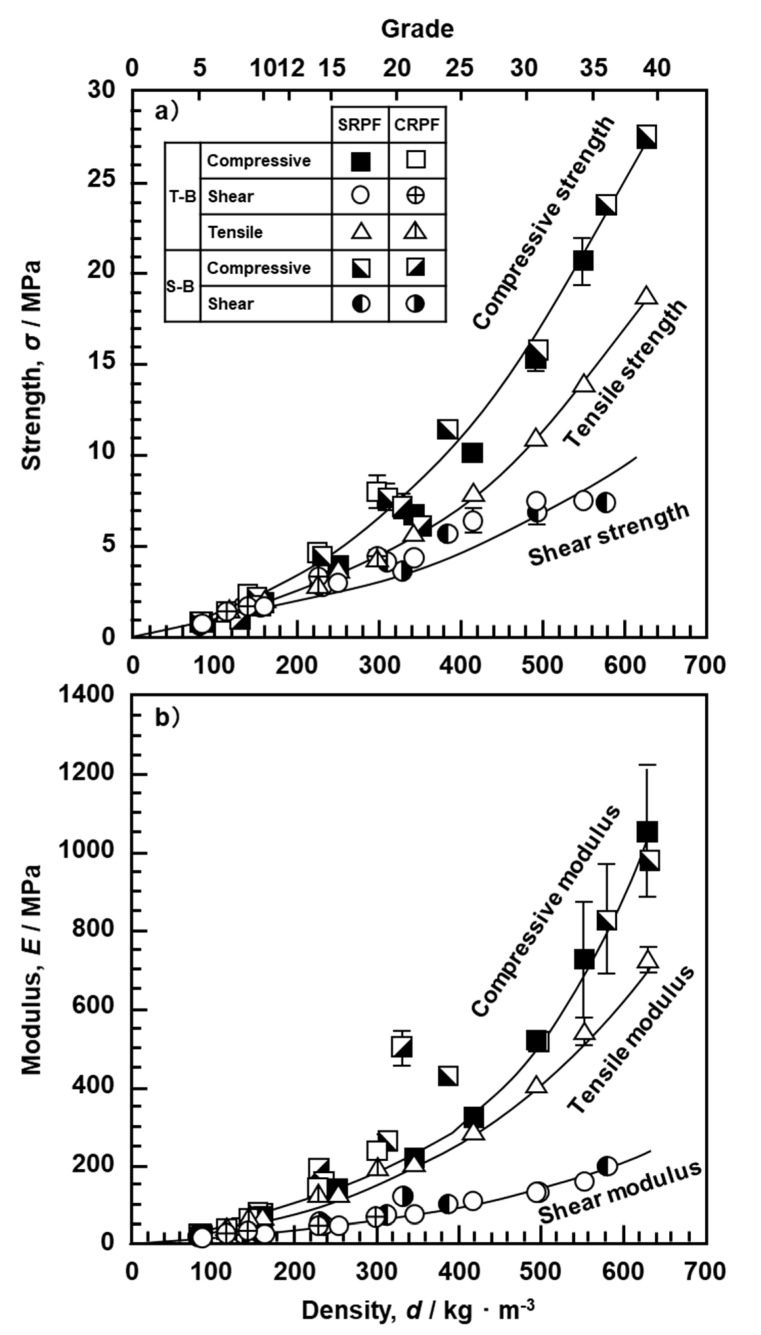
Effect of density on strength and modulus obtained by compressive, shear, and tensile tests with T- and S-bones. (**a**) Strength and (**b**) Modulus.

**Figure 9 materials-13-04836-f009:**
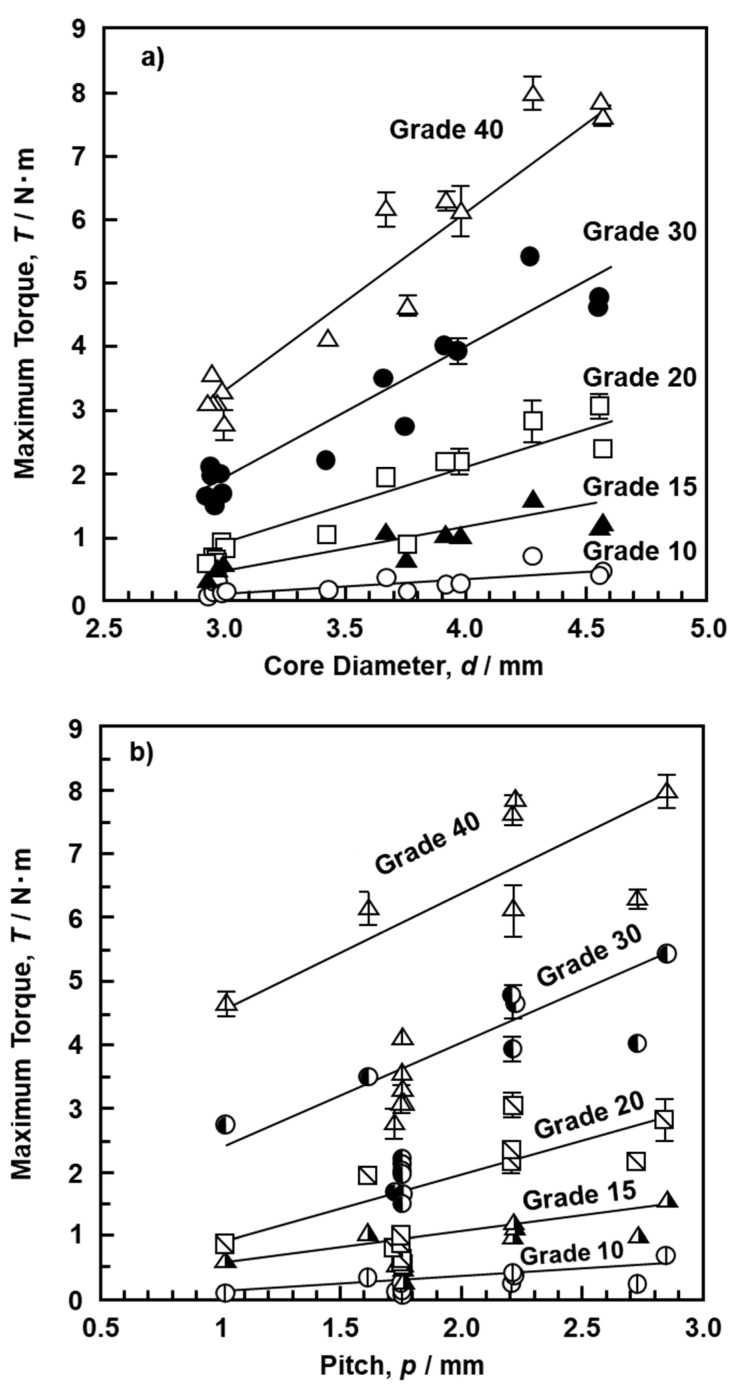
Effects of (**a**) core diameter and (**b**) pitch on maximum driving torque with cortical screws.

**Figure 10 materials-13-04836-f010:**
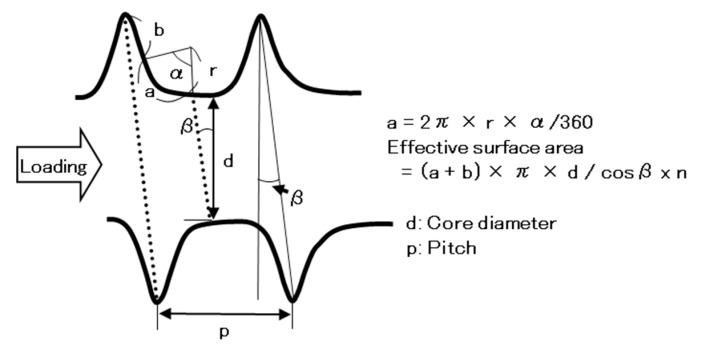
Schematic illustration of effective surface area of the screw.

**Figure 11 materials-13-04836-f011:**
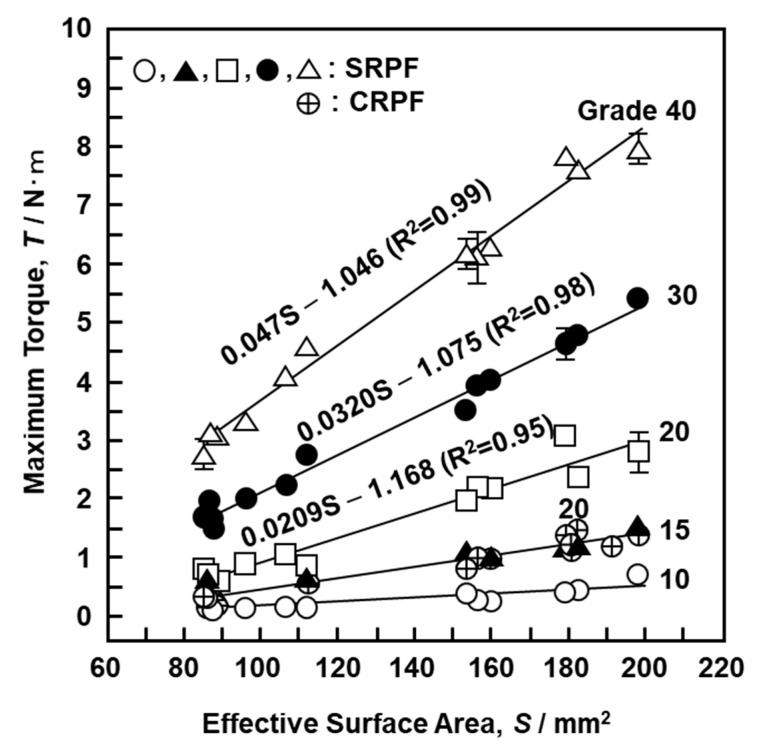
Effect of effective surface area of the screw on maximum driving torque.

**Figure 12 materials-13-04836-f012:**
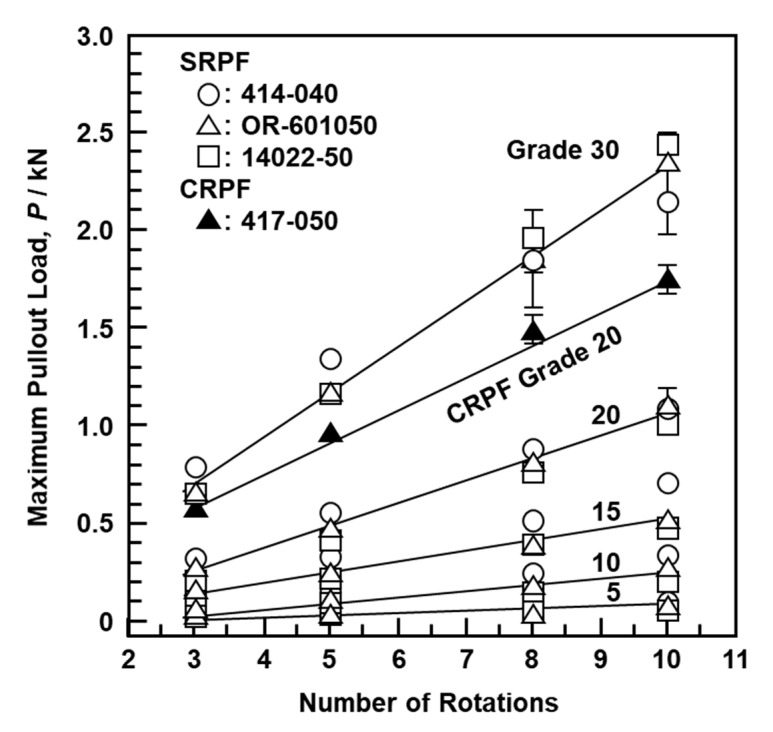
Effect of number of rotations on maximum pullout load with S-bone SRPF and grade 20 CRPF.

**Figure 13 materials-13-04836-f013:**
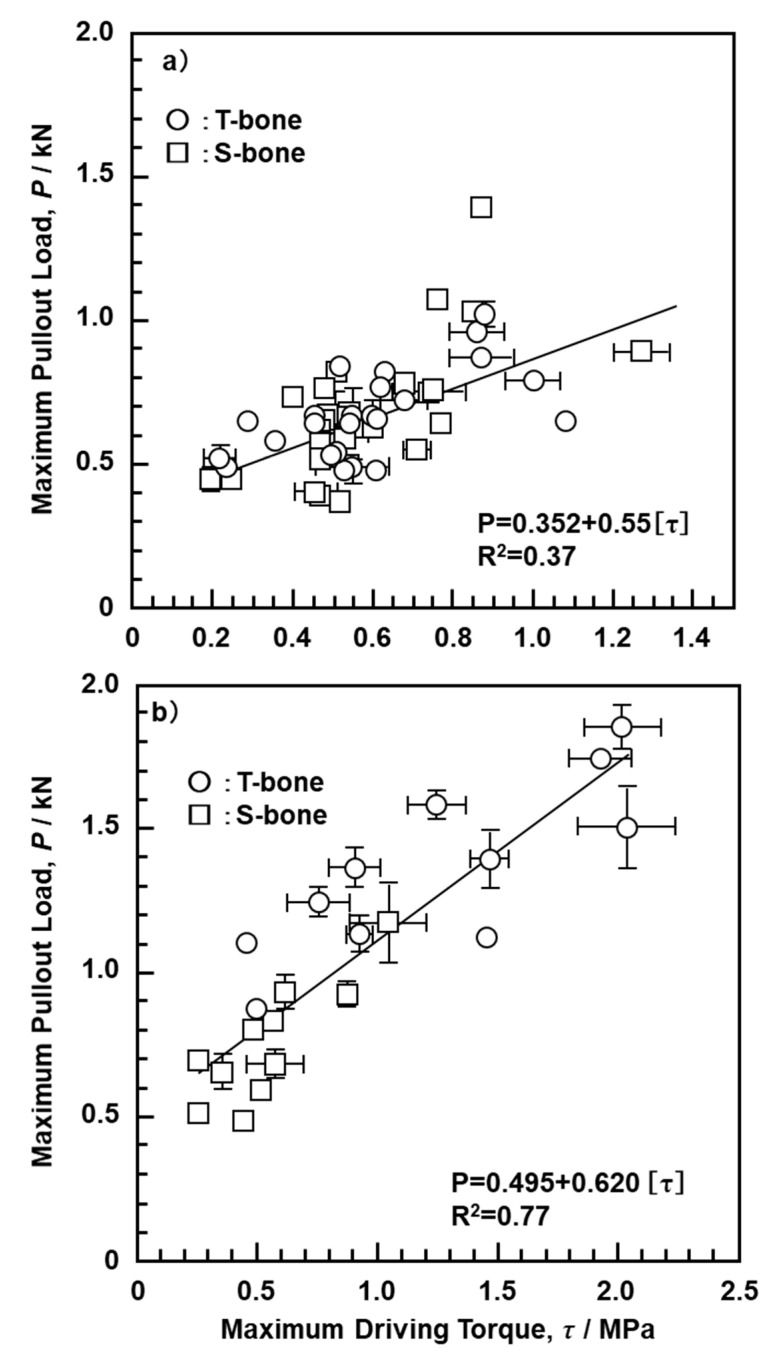
Relationship between maximum pullout load and maximum driving torque with (**a**) SRPF grade 15 and (**b**) CRPF grade 20.

**Figure 14 materials-13-04836-f014:**
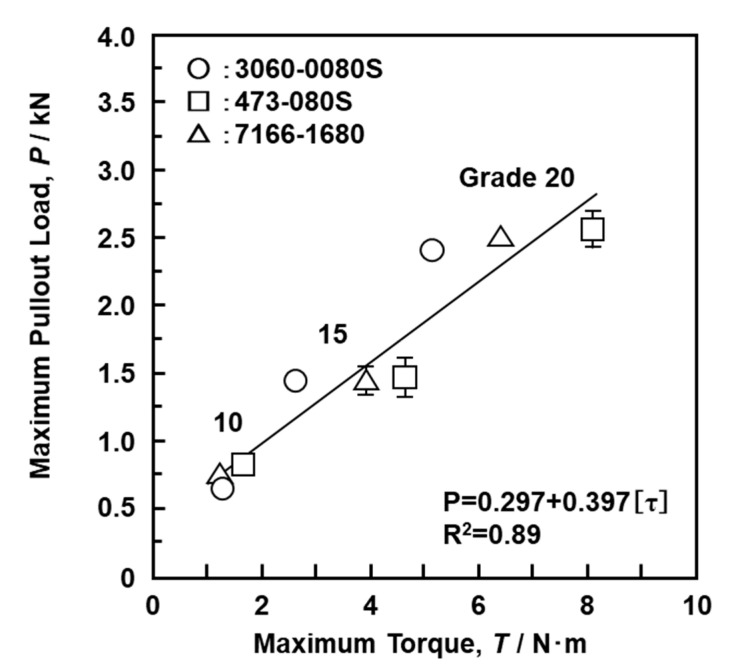
Relationship between maximum pullout load and maximum driving torque with S-bone SRPF obtained using lag screws (80 mm) of short femoral nails.

**Table 1 materials-13-04836-t001:** Various orthopedic screws used in this study.

Supplier		Length (mm)	Length of Threaded Portion (mm)	Thread dia. (mm)	Core dia. (mm)	Pitch (mm)
(1) Self-tapping fully threaded cortical screws						
(i) Depuy Synthes						
C.P. Ti	414-040	40	35	4.5	3.0	1.75
	414-850S	50	45	4.4	3.0	1.75
Ti-6-4	480-990VS	36	26	3.9	3.0	0.70
Stainless	214-850	50	45	4.4	3.0	1.75
	280-990	36	26	3.9	3.0	0.70
(ii) Zimmer Biomet						
Ti-6-4	48-2319-038-00	38	33	4.6	3.8	1.02
	48-2319-050-00	50	45	4.6	3.8	1.02
	48-2306-050-01	50	43	4.4	3.0	1.72
Stainless	00-2253-050-42	50	45	4.2	3.7	1.61
	00-2253-050-55	50	45	5.5	5.0	1.60
(iii) Mizuho						
Ti-6-4	01-810-34	34	29	4.5	3.0	1.75
	01-810-36	36	31	4.5	3.0	1.75
	01-810-38	38	32	4.5	3.0	1.75
	01-810-46	46	41	4.5	3.0	1.75
	01-810-48	48	43	4.5	3.0	1.75
Ti-Zr	01-810-46	46	41	4.5	3.0	1.75
(iv) Stryker						
Ti-6-4	OR-601050	50	44	4.5	2.9	1.75
	1896-5050S	50	46	5.0	3.9	1.75
(v) MDM						
Ti-6-4	14022-50	50	45	4.6	3.7	1.00
	14224-50	50	43	5.0	3.6	1.83
(vi) Teijin Nakashima Medical						
Ti-6-4	B30	50	45	4.5	2.9	1.75
	B35	50	45	4.5	3.4	1.75
(vii) Meira						
Ti-6-4	035A-001-050	50	45	3.6	2.5	1.24
(2) Cannulated cancellous screw (partially threaded)						
(i) Depuy Synthes						
C.P. Ti	417-050	50	32	6.5	3.1	2.76
Stainless	217-050	50	32	6.5	3.0	2.75
(ii) Zimmer Biomet						
Ti-6-4	47-2483-095-60	95	32	6.0	4.7	1.72
(iii) MDM						
Ti-6-4	14225-50	50	20	5.0	3.6	1.73
(iv) Teijin Nakashima Medical						
Ti-6-4	B050	50	30	6.0	4.3	1.76
(v) Meira						
Ti-6-4	005A-340-070	70	25	4.2	2.8	1.50
(3) Locking bolts used in intramedullary femoral nails						
Depuy Synthes						
Ti-6-4	459-300VS	30	25	4.8	4.2	2.78
Ti-6-4	459-500VS	50	45	4.8	4.2	2.78
(4) Pedicle screws						
(i) Robert Reid						
Ti-6-4	ISOLA 2226-2440	45	38	5.5	4.0	2.21
	ISOLA 2226-2840	45	38	6.3	4.6	2.21
	ISOLA 2230-07R	55	48	6.3	4.6	2.22
(ii) Medtronic						
Ti-6-4	8695540	45	40	5.6	3.9	2.73
	86946540	45	40	6.6	4.3	2.85
(iii) Zimmer Biomet						
Ti-6-4	3306-4540	40	32	4.5	3.2	2.37
	3306-5540	40	32	5.5	4.0	2.54

**Table 2 materials-13-04836-t002:** Bone models used in this study.

Solid Rigid Polyurethane Foam (SRPF)								
Grade	5	10	15	20	25	30	35	40
Nominal density (kg/m^3^)	80.1	160.2	240.3	320.4	400.5	480.6	560.6	640.8
Cellular rigid polyurethane foam (CRPF)								
Grade	7.5	10	15	20	−	−	−	−
Nominal density (kg/m^3^)	120.15	160.2	240.3	320.4	−	−	−	−

**Table 3 materials-13-04836-t003:** Mechanical properties obtained by torsional breaking and four-point bending durability tests with various screws.

Material	Torsional			Durability	Tensile [1]		
	τ_2°_/MPa	τ_max_/MPa	G/MPa	σ_D_/MPa	σ_0.2_/MPa	σ_UTS_/MPa	σ_D_/σ_UTS_ (%)
(1) Full-thread cortical screws							
C.P.Ti	552	948	1616	498	568	725	69
Ti-6-4	650 ± 95	986 ± 67	2140 ± 50	804 ± 77	892	990	82
Stainless	824 ± 5	1012 ± 53	1421 ± 80	732 ± 133	876	1051	70
Ti-Zr	598	870	2698	-	-	-	-
(2) Cannulated cancellous screws							
C.P.Ti	849	1259	4889	-	-	-	-
Ti-6-4	535 ± 2	730	-	-	-	-	-
Stainless	1013	1304	-	-	-	-	-
(3) Locking bolt							
Ti-6-4	535	776	3254	-	-	-	-

**Table 4 materials-13-04836-t004:** Mechanical properties (minimum and maximum requirements) of bone models used for testing orthopedic devices.

Grade	Density/kg·m^−3^	Compressive		Shear	
		Strength/MPa	Modulus/MPa	Strength/MPa	Modulus/MPa
5	72−88	0.4−1.0	12−28	0.4−0.9	5.5−10
10	144−176	1.7−2.8	46−78	1.2−2.0	15−30
12	173−212	2.5−4.0	65−100	1.6−2.6	20−40
15	216−265	3.8−6.1	98−170	2.2−3.5	27−60
20	289−353	6.5−10	170−270	3.4−5.4	40−90
25	361−441	10−16	250−390	4.0−7.3	56−130
30	433−529	14−23	360−550	5.0−9.5	72−200
35	505−617	18−31	470−800	7−12	90−250
40	577−705	25−40	600−1100	8−15	110−300

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
