# Peer review of "Mechanical Performance of Metallic Bone Screws Evaluated Using Bone Models"

_materials, 2020, doi:10.3390/ma13214836_

Round 1

Reviewer 1 Report

This article is very clearly structured, well-written, and with clinical relevance. This study investigated the mechanical properties of different types of the cortical bone screw, 13 cancellous bone screw, and locking bolt. The results provide significant information about the design of screw for orthopaedic surgery. The authors have discussed the topic in detail. I suggest that the limitation and outlook of this study can be further explained. Afterwards, the manuscript can be recommended for publication.

Author Response

Thank you for the peer review of our manuscript.

Reviewer 2 Report

The paper deals with mechanical testing of many metallic bone screws used in orthopaedical surgery with the aid of bone models (solid rigid polyurethane foam, and cellular rigid polyurethane foam) having different nominal density. It is well written, there is an impressive amount of results. However, in the Experimental method paragraph, all the bolts are listed one after another (line 108-147). This is very confusing to the reader who cannot easily orient himself in the text. The same is valid for listing of bone models (line 149-155). Both problems could be resolved inserting tables instead (e.g. in the attached *.pdf file as a suggestion). After replacing lists of bolts by tables, the paper will be somewhat longer, but much better arranged for the readers. So I recommend the paper for publication after this minor, but mandatory correction.

Author Response

Thank you for the peer review of the manuscript, which has been revised in accordance with your comments. Following the reviewers’ suggestion, the screws and bone models have been listed in the Tables 1 and 2.

Reviewer 3 Report

Complex bone fractures requiring surgical intervention are common in the era of advanced communication and competitive sports. The literature - especially clinical - describes cases of bone fixation with plates and screws, which are designed to keep the bone fragments in a fixed position and allow them to heal. However, there are also reports in the literature that such screws do not fulfill their role - they loosen, corrode, often break, thus leading to the need to perform the operation again. Previous studies - as indicated by the authors of the study - concerned mainly bending and compression tests of the elements that fix the bone, but there are only a few sources indicating studies carried out on the bone models used by the authors. Therefore, I believe that this work is devoted to a very important, current and necessary topic - the study of the influence of the core diameter on the torsional fracture properties and durability of metal bone screws, the relationship between the moment and the pull-out strength of bone screws, and the relationship between density and mechanical properties bone models. The research and analytical process is complete and includes all possible stages, specified by the authors in the methodology. The introduction is concise, but contains the information necessary to become familiar with the topic. The work methodology is extensive and described in detail. The research methodology is impeccable. Noteworthy is the large number of experimental studies conducted on the bone model. However, in the conclusions, I do not refer to the material that turned out to be the best, which showed the best parameters.

Despite a positive assessment, I have to point out a few aspects that need to be supplemented:

  1. The authors use the names Ti-6Al-4V and Ti-6-4 interchangeably in many places - I think it would be appropriate to use the same nomenclature.
  2. Figure 5. Effects of core diameter of screw on (a) 2° proof torque, (b) rupture torque, and (c) rupture angle  - Ti-Zr is marked in the drawing, but not in the table below.
  3. The signature for Fig 9 requires formatting
  4. Fig 9 - legend is missing.
  5. In the methodology and description of the research, the names of the materials used are repeated – I think it is unnecessary.

Author Response

Thank you for the peer review of the manuscript, which has been revised in accordance with your comments. The modifications are as follows:

(1) We have corrected the notation to Ti-6-4.

(2) The mark description and the mechanical properties for Ti-Zr alloy have been added to Fig. 5 and Table 3, respectively.

(3), (4) Fig. 9 has been modified along with its caption.

(5) The material type has been deleted. However, we have left the catalog numbers to make it easier for the reader to understand which screws are being tested.